genomics, ecology, molecular biology

ocean acidification, climate change, parental effects, phenotypic variation, transcriptome, genetic variance

**Author for correspondence:**
Timothy Ravasi
e-mail: timothy.ravasi@oist.jp

# Molecular basis of parental contributions to the behavioural tolerance of elevated pCO$_2$ in a coral reef fish

Alison A. Monroe[1,2], Celia Schunter[3], Megan J. Welch[4], Philip L. Munday[4] and Timothy Ravasi[4,5]

[1]Division of Biological and Environmental Science and Engineering, King Abdullah University of Science and Technology, Thuwal, Saudi Arabia
[2]Marine Genomics Laboratory, Department of Life Sciences, Texas A&M University Corpus Christi, Corpus Christi, TX 78412, USA
[3]Swire Institute of Marine Science, The School of Biological Sciences, The University of Hong Kong, Pokfulam Road, Hong Kong SAR
[4]Australian Research Council Centre of Excellence for Coral Reef Studies, James Cook University, Townsville, Australia
[5]Marine Climate Change Unit, Okinawa Institute of Science and Technology Graduate University (OIST), 1919-1 Tancha, Onna-son, Okinawa 904-0495, Japan

AAM, 0000-0002-9372-4889; CS, 0000-0003-3620-2731; PLM, 0000-0001-9725-2498; TR, 0000-0002-9950-465X

Knowledge of adaptive potential is crucial to predicting the impacts of ocean acidification (OA) on marine organisms. In the spiny damselfish, *Acanthochromis polyacanthus*, individual variation in behavioural tolerance to elevated pCO$_2$ has been observed and is associated with offspring gene expression patterns in the brain. However, the maternal and paternal contributions of this variation are unknown. To investigate parental influence of behavioural pCO$_2$ tolerance, we crossed pCO$_2$-tolerant fathers with pCO$_2$-sensitive mothers and vice versa, reared their offspring at control and elevated pCO$_2$ levels, and compared the juveniles' brain transcriptional programme. We identified a large influence of parental phenotype on expression patterns of offspring, irrespective of environmental conditions. Circadian rhythm genes, associated with a tolerant parental phenotype, were uniquely expressed in tolerant mother offspring, while tolerant fathers had a greater role in expression of genes associated with histone binding. Expression changes in genes associated with neural plasticity were identified in both offspring types: the maternal line had a greater effect on genes related to neuron growth while paternal influence impacted the expression of synaptic development genes. Our results confirm cellular mechanisms involved in responses to varying lengths of OA exposure, while highlighting the parental phenotype's influence on offspring molecular phenotype.

## 1. Introduction

As atmospheric CO$_2$ levels increase, so does the amount of CO$_2$ taken up by the ocean, causing a decrease in seawater pH (ocean acidification (OA)), with potentially broad-ranging effects on the physiology and ecology of marine organisms [1]. However, the effects of OA on marine ecosystems will depend on the relative sensitivity and tolerance of different species, and their ability to acclimatize and adapt to the environmental changes brought about by rising CO$_2$ levels [2]. Recent studies show that increased pCO$_2$ levels can affect the growth, survivorship and physiology of some marine fishes [3–7]. Elevated CO$_2$ has also been shown to alter behaviours in a wide variety of fish [8–10] and invertebrates [11–13]. However, most experimental studies

focus on short-term exposure to elevated $pCO_2$ and do not account for phenotypic variation that may enable populations to adapt over the time scale at which OA will occur [2,14].

Fish use chemical alarm cues (CACs) from conspecifics to detect predation threats and respond to CAC presence by moving away from the cue and reducing activity [15]. Under elevated $pCO_2$ conditions, juvenile fish can exhibit an impaired response to CAC, showing a decreased avoidance to CAC, and in some cases failing to associate CAC with the threat of predation [16–19]. The underlying cause of this behavioural change is thought to be impaired function of $GABA_A$ receptors (GABA is the major inhibitory neurotransmitter in the brain) caused by changes in the concentration of acid-base relevant ions to maintain pH homeostasis in a high $CO_2$ environment [17,20–22]. Elevated $CO_2$ has also been demonstrated to affect learning ability, decision making, turning preference, auditory preferences, visual acuity, shoaling, boldness and escape responses in a range of different fish species [16,18,23–25]. Such alterations in behaviour could affect individual performance and survivorship, with potential implications for community structure and population replenishment [8,26]. However, there is also individual variation in behavioural sensitivity to elevated $CO_2$, with some individuals exhibiting impaired behaviours, whereas others do not, especially at $CO_2$ levels projected to occur in the ocean this century (i.e. approximately 700 µatm $CO_2$) [23,27,28]. Such individual variation could be the raw material for population-level adaptation to rising $CO_2$ levels [14,29].

Genetic variation among parents, combined with non-genetic effects from the parental environment, drive an offspring's phenotype through natural selection, epigenetic inheritance and changes in molecular mechanisms [30]. These parental effects, genetic and non-genetic, are key to understanding how populations will adapt and survive in the face of climate change. Both mothers and fathers make important contributions to the success and development of their offspring, yet recent studies have focused singularly on either maternal or paternal effects, rarely examining how they may interact to shape their offspring's performance [31]. In the marine stickleback, maternal exposure to high temperatures had a clear impact on the growth of their offspring in the same environmental conditions [32]. Maternal inheritance of increased metabolic capacity led to larger juveniles, suggesting transgenerational plasticity can mediate short-term impacts of ocean warming [32]. Another recent study, on wild salmon, found a correlation between telomere length in juveniles and paternal time spent in seawater that appeared unaffected by increased temperatures [33]. Short telomeres can indicate poor biological health, suggesting that fathers with an increased time at sea produce healthier offspring even in the face of environmental variability [33]. There is a significant lack of knowledge about sex-specific parental contributions in fish, especially under elevated $pCO_2$ conditions, and the interactions of genetic and non-genetic effects add to the complexity [34]. Identifying the parental contributions to specific traits and the molecular mechanisms behind them will allow us to better predict how fish will respond and adapt to rapid climate change.

Recent studies on the spiny damselfish, *Acanthochromis polyacanthus,* indicate that individual variation in sensitivity of the behavioural reaction to elevated $pCO_2$ is heritable and could thus facilitate adaptation [28,35,36]. A portion of fish from the natural population respond normally to CACs under predicted end-of-century $pCO_2$ levels of 700–800 µatm and thus appear to be tolerant to the behavioural effects of OA. Moreover, there is a strong correlation between the behavioural tolerance of juvenile spiny damselfish to elevated $pCO_2$ and the tolerance of their fathers in both wild and captive populations [28], indicating that behavioural tolerance to elevated $pCO_2$ has a genetic basis and is heritable. Furthermore, molecular differences were identified between offspring of tolerant versus sensitive parental pairs, with those under transgenerational exposure to elevated $pCO_2$ showing differential gene and protein expression in the brain [35,37]. Specific genes related to this signature of tolerance include those that regulate the circadian rhythm processes and could be creating a phase shift in the circadian clock to better control their acid-base regulation [35]. Recent evidence suggests that this behavioural variation may be passed on paternally through generations [28]; however, it is still unknown if this heritability is identifiable in transcriptional changes.

In this study, we investigate maternal and paternal influence on the molecular signature of behavioural tolerance to elevated $pCO_2$ in *A. polyacanthus*. Using a unique transgenerational experimental design, we cross-bred behaviourally tolerant mothers with behaviorally sensitive fathers and behaviourally sensitive mothers with tolerant fathers. We exposed breeding pairs and their offspring to ambient and elevated $pCO_2$ conditions. As described in previous studies [28,35,36], offspring were exposed to ambient $pCO_2$ (control), elevated $pCO_2$ from hatching (transgenerational high $pCO_2$ treatment), or for only 4 days before sampling (acute high $pCO_2$ treatments). Due to previous research implicating the role of neurotransmitters in behavioural changes under elevated $pCO_2$, we continued to study the brain. We measured genome-wide gene expression in 68 juvenile fish from the different combinations of parental pairs ($CO_2$-tolerant fathers paired with $CO_2$-sensitive mothers versus $CO_2$-sensitive fathers paired with $CO_2$-tolerant mothers) and $CO_2$ treatment conditions, exploring the role of the maternal and paternal phenotype in influencing individual offspring expression profiles under various $pCO_2$ levels and exposure durations.

## 2. Material and methods

### (a) Experimental design

In this study, we sampled the brains of juvenile *A. polyacanthus* from the laboratory experiment described in a previous paper [28]. Full details of the experimental design are provided in that paper [28]. Briefly, adult *A. polyacanthus* were collected from the northern Great Barrier Reef in Australia and transported to the aquaria facilities at James Cook University (Townsville, Australia; JCU ethics permit A1828). Adults were placed in elevated $pCO_2$ (754 µatm) (electronic supplementary material, table S1) for 7 days then tested for their reaction to CAC in a two-channel flume. Their sensitivity to high $pCO_2$ was determined by the amount of time spent in the CAC: spending less than 30% of the time in the CAC was considered tolerant to elevated $pCO_2$, and individuals spending greater than 50% were considered non-tolerant. Adults were then sexed and paired for breeding based on their behavioural sensitivity. For the purpose of this study, we focused only on the tolerant male × sensitive female (T♂S♀) and sensitive male × tolerant female (S♂T♀) pairs; however, all parental combinations were made for the larger experimental design [28,36]. Adults were then placed in 40 L aquaria at control (414 µatm) and elevated $pCO_2$

**Figure 1.** Experimental design detailing the creation of breeding pairs, the holding conditions for parents and offspring, the testing conditions prior to euthanasia for the offspring and the names of the four different treatments as they are identified in the text. The blue colour refers to tolerant female, sensitive male parental pairs while green indicates tolerant male, sensitive female parental pairs.

(754 µatm) conditions for three months prior to breeding season (electronic supplementary material). For this specific study, we used clutches from 3T♂S♀ pairs and 3 S♂T♀ pairs in control and 2T♂S♀ pairs and 2 S♂T♀ pairs at elevated $pCO_2$. Breeding pairs were checked daily for the presence of egg clutches. Consistent with previous observations for this species, there was no apparent effect of elevated $pCO_2$ on reproductive output, although this was not directly quantified in the current study. On the day of hatching (approx. 10 days post-fertilization), the offspring were transferred to either control or elevated $pCO_2$ conditions (figure 1). This led to two long-term conditions referred to as control (control parents, offspring reared in control $pCO_2$) and transgenerational (elevated $pCO_2$ parents, offspring reared in elevated $pCO_2$). Furthermore, we exposed half of the offspring from the control condition to elevated $pCO_2$ (754 µatm) 4 days prior to sampling at the end of the experiment. These offspring came from parents exposed to either control conditions or elevated $pCO_2$ creating two separate offspring acute treatments named for their parental conditions: control-acute and high $pCO_2$-acute (figure 1). No significant juvenile mortality was recorded throughout the experiment across all conditions. After five months, up to four fish per parental pair were euthanized resulting in approximately nine individuals from each parental type in each of the four conditions for a total of 68 fish ($n = 16$ or 18 from each of the four conditions, electronic supplementary material, table S2). Brains were dissected out and immediately flash frozen in liquid nitrogen and then kept at −80°C for further processing. Different fish were euthanized for this study than those used by Welch & Munday [28] to measure behavioural response to CAC. In the previous study, offspring were tested after six weeks in a two-channel flume to determine their behavioural phenotype [28]. For this study, we sampled fish after five months to ensure sufficient brain tissue was available for the molecular analysis. We did not perform any behavioural testing prior to sampling at five months to avoid potential influence of handling disturbance and the testing regime on the brain transcriptome.

## (b) RNA extraction and preparation

Whole-frozen fish brains were homogenized in RLT-plus buffer for 30 s using a bead beater (BioSpec) with single-use silicone beads and total RNA was extracted using the AllPrep DNA/ RNA mini kit following the provided protocol (Qiagen). RNA quantity was determined using a Nanodrop (Thermo Scientific) then reanalysed for both the quality and quantity with the Agilent 2100 Bioanalyzer. Extracted RNA was processed at the King Abdullah University of Science and Technology (KAUST) Bioscience Core Lab for library preparation and sequencing. Conversion to cDNA and preparation for Illumina sequencing was done using the TruSeq RNA Illumina Library Prep Kit. Samples were sequenced on the Illumina HiSeq 2500 paired end at 100 bp in KAUST, Saudi Arabia.

## (c) Gene expression analysis

Raw reads were quality checked with FastQC [38] and sequences were trimmed using Trimmomatic v. 0.36 [39] with set parameters including a phred score of 33, a minimum length of 40 and removal of the first 13 bp of each sequence as well as the Illumina adapter sequences. Reads were rechecked in FastQC for quality to verify correct adapter trimming. Sequences were then aligned to the *Acanthochromis polyacanthus* genome (ENSEMBL ASM210954v1) using hisat2 v. 2.1.0 [40] with an average successful alignment rate of approximately 80%. SAM output files from mapping were then sorted and converted into bam files using SAMtools v. 1.5. Feature counts (Subreads v. 1.5.3) [41] was used to create a matrix of raw exon read counts and the gene-ids provided in the genome annotation. Minimum mapping quality was set to 20 and both fragment pairs were required to align to the reference. The data matrix was then imported into R v. 3.4.0 and differential expression between conditions and parental pairs were statistically determined with DESeq2 [42]. A likelihood ratio test (LRT) was used to determine the interactions between the multiple variables as well as the best design formula for further analysis ($p$-adj < 0.05). Significant differentially expressed genes (DEGs) were identified using a $q$-value < 0.05 and a minimum log twofold change of 0.3. All graphs were created using the R packages Vegan and ggplot2 [43,44].

Gene ontology of each gene was annotated using Blast2Go Pro v. 4.19 (Gene Ontology Consortium [45]. Enrichment was performed in RStudio v. 1.1.463 with the GO-MWU script that uses a Mann–Whitney $U$ test and a Benjamini–Hochberg correction to determine statistical significance [46], (scripts available at: https://github.com/z0on/GO_MWU). The enrichment test was carried out separately for each of the different GO domains using stringent filtering including collapsing redundant categories, only selecting those genes represented by more than 5 GO-Terms, and employing an FDR cutoff of 0.1.

## 3. Results

## (a) Condition-specific expression patterns

Duration of exposure to elevated $pCO_2$ had the greatest effect on expression profiles of *A. polyacanthus* offspring. The number of DEGs was higher when comparing the two acute-elevated $pCO_2$ treatments to control than in the comparison of the transgenerational-elevated $pCO_2$ treatment to control (figures 2 and 3). The influence of length of exposure to elevated $pCO_2$ on patterns of gene expression in the brain was also seen by the high number of overlapping DEGs between offspring of both acute treatments (1642), whereas offspring

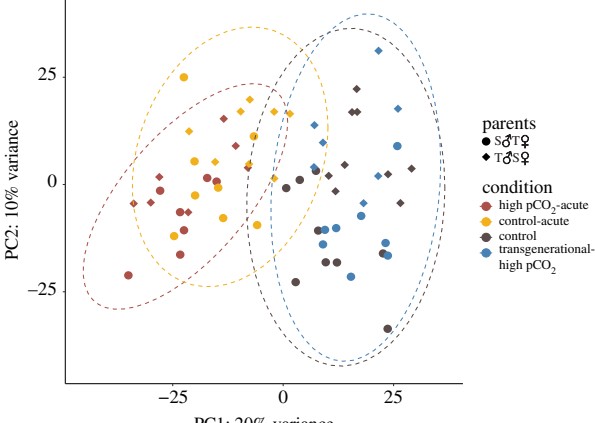

**Figure 2.** Principal coordinate analysis of variance stabilized read counts. Parental identity is indicated by shape and condition by colour. Ellipses designate clusters by condition and are coloured to match. (Online version in colour.)

from the transgenerational treatment only shared one DEG with the acute treatments. When comparing the control and transgenerational treatments, there is a stronger grouping by parental identity than by condition, but this grouping is less evident within the two acute treatments (figure 4a,b).

## (b) Influence of parental identity in control and transgenerational treatments

Comparing offspring from tolerant mothers (S♂T♀ parental pair) with offspring from tolerant fathers (T♂S♀ parental pair), we identified 1592 and 1443 significant DEGs in the control $pCO_2$ and transgenerational-elevated $pCO_2$ treatments, respectively (figure 3). This revealed a clear influence of parental phenotype on the expression profile of the offspring when fish were kept transgenerationally at the same $pCO_2$ levels (figure 4a). Functionally enriched pathways included an upregulation of neuropeptide activity, developmental growth and dopamine receptor signalling in offspring of tolerant fathers in both control and the transgenerational-elevated $pCO_2$ treatment. By comparison, pathways such as spectrin binding, mismatch repair and RNA splicing were identified as significantly upregulated in offspring of tolerant mothers under control and transgenerational-elevated $pCO_2$.

There were 264 DEGs when comparing different environmental $pCO_2$ levels (control versus transgenerational-elevated $pCO_2$) for offspring of tolerant mothers, but only 68 DEGs when comparing these treatments for offspring of tolerant fathers (figure 3). Only 11 of these genes were shared between offspring of the two parental groups, including upregulation of a gene known to protect cells from reactive oxygen species (*GSTA1*) and a cytoskeleton-related gene (*CKAP2*). Genes significantly differentially expressed in the offspring of tolerant mothers included several related to growth (*TNNT3*, *MYSS* and *MLRV*). Further analysis on total body weight revealed that offspring of tolerant mothers were on average smaller than those of tolerant fathers, with a significant difference between the two in the transgenerational-elevated $pCO_2$ treatment (electronic supplementary material, figure S1 and table S3).

GO enrichment revealed upregulation of the circadian rhythm pathway as well as circadian regulation of gene expression in offspring from tolerant mothers, including the circadian repressors *PER3*, *CIART* and *HT7R* (figure 4). Unique differentially enriched pathways in offspring of

tolerant fathers when comparing control to transgenerational included downregulation of reactive oxygen species biosynthesis and the H4 histone complex as well as upregulation of axon development (figure 5).

## (c) Influence of parental identity in acute-elevated $CO_2$ treatments

Fewer DEGs due to parental identity were identified in both acute-elevated $pCO_2$ treatments compared with the control and transgenerational-elevated $pCO_2$ treatment. Two hundred and seventy-two DEGs were found between offspring of T♂S♀ (tolerant fathers) and S♂T♀ (tolerant mothers) parents in the control-acute treatment, and 270 DEGs between offspring of the two parental pairs in the high $pCO_2$-acute treatment (figure 3). There was a strong common response to the acute-elevated $pCO_2$ treatments versus control, regardless of the parental condition or parental identity, which consisted of 1642 DEGs. The enriched functions for these shared genes include collagen binding, neurotransmitter receptor activity and GABA receptor activity (figure 5). GABA-related genes included *KCC1*, *NXPH1* and *NXPH2*, several solute carriers specializing in GABA transport, and *GAD2* involved in GABA catalysis. Carbonic anhydrase (CA) genes (*CAH4*, *8*, *10* and *15*) were also differentially expressed between control and both acute-elevated $pCO_2$ conditions. Circadian rhythm activators (*RORa* and *b*) involved in stability of the clock were highly upregulated in all acute treatments when compared to control.

Comparing control to control-acute and high $pCO_2$-acute led us to identify 604 shared DEGs from offspring of tolerant fathers and 567 shared DEGs from offspring of tolerant mothers. In offspring of tolerant mothers, *SC6A1* and *S6A11* were both upregulated in the acute-elevated $pCO_2$ treatment. Offspring of tolerant mothers exhibited highly upregulated glutamate-related genes (*NMD3B* and *GRIK2*), as well as genes related to neuron plasticity, growth and development (*AMIGO1*, *GFRA2* and *CDK5R1*). Several genes downregulated in offspring of tolerant mothers exposed to acute-elevated $pCO_2$ conditions were related to development and growth (*TNNI2*, *TNNC2*, *MYSS*, *ACTA1* and *ACTN3*). Pathways unique to the offspring of tolerant mothers were nervous system development, serine family biosynthesis and RNA methylation (figure 5). In offspring of tolerant fathers, solute carrier genes involved in GABA transport were both upregulated (*S6A11* and *S6A13*) and downregulated (*SC6A1*) in acute-elevated $pCO_2$ conditions. These offspring also showed upregulation of genes related to synaptic plasticity (*MPDZ*) and inhibition of synaptic development (*IGSF9B*) as well as a cytoskeleton gene (*ANK1*) and growth factor genes (*FGF6* and *FGF14*) involved in cell proliferation and nervous system development. Downregulated genes including transcriptional repressors (*HES5* and *6*) as well as *NEUROG1* involved in transcriptional regulation and cell differentiation were identified in offspring of tolerant fathers under acute-elevated $pCO_2$ conditions. Also identified as downregulated genes were a histone-binding gene (*NASP*), a gene involved in immunity and cell death (*PERF*), and one involved in DNA repair (*PAF15*). These genes corresponded to unique pathways such as histone binding, ATPase activity, hydrogen transport and microtubule polymerization enriched in offspring of tolerant fathers (figure 5).

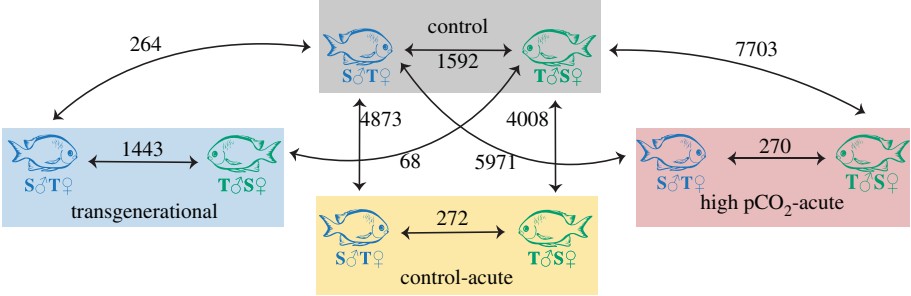

**Figure 3.** Number of DEGs from pairwise comparisons between parental pairs within conditons and between conditions with parental pairs as a factor. Fish colour indicates parental identity while rectangle colour indicates labelled offspring treatment. (Online version in colour.)

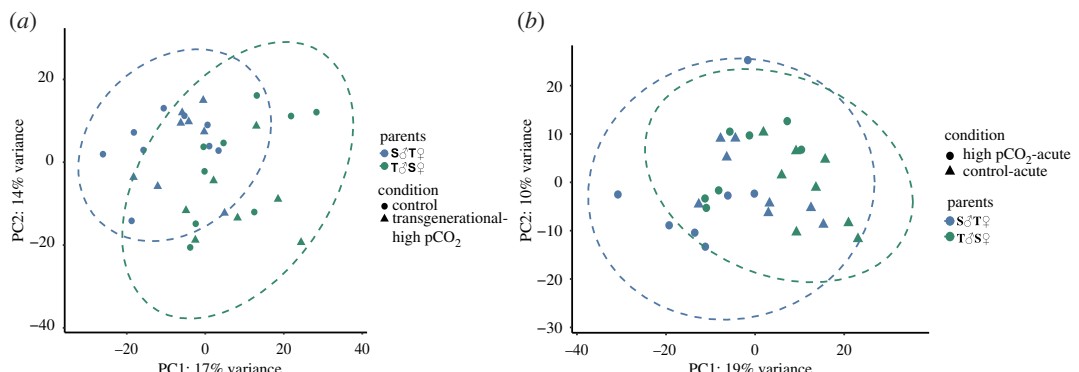

**Figure 4.** Principal component analysis of the top 5000 significant genes after running the LRT; (*a*) displays offspring from chronic treatments while (*b*) shows those from acute treatments. Condition is indicated by shape and parental identity by colour. Ellipses signify clusters due to parental identity and are coloured to match. (Online version in colour.)

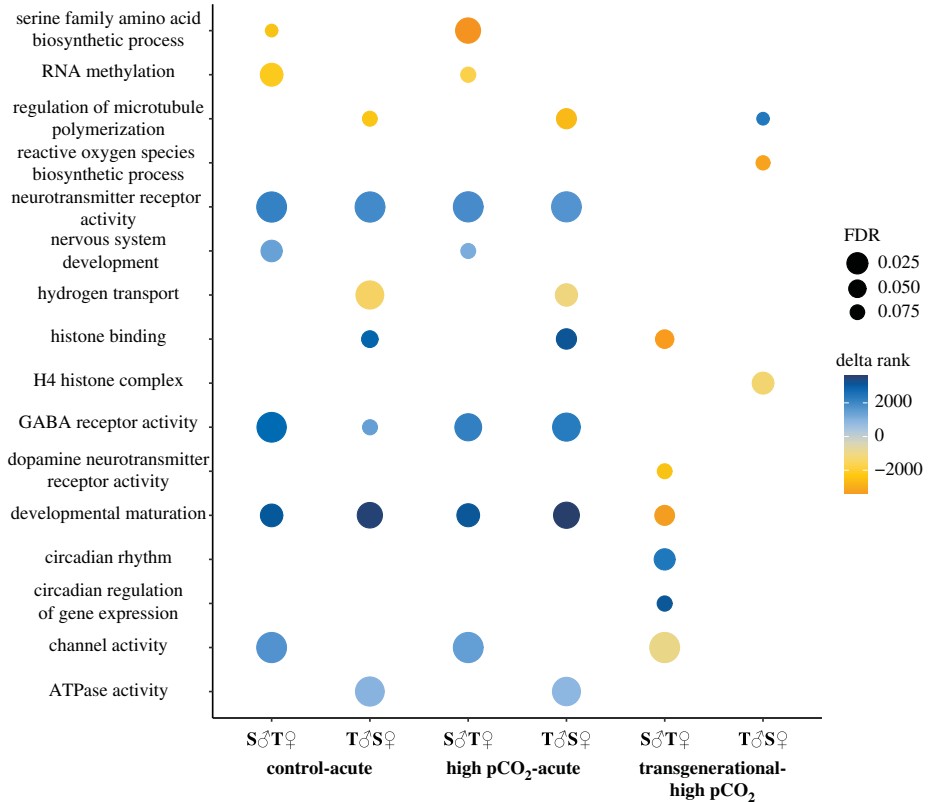

**Figure 5.** Graph of significantly enriched GO terms between all conditions compared to control, within parental pairs. Colour indicates the delta rank statistic from the GO-MWU enrichment while bubble size indicates FDR. GO terms associated with upregulated genes are considered positive while those related to downregulated genes are negative. (Online version in colour.)

# 4. Discussion

In this study, we show that parental phenotype shapes the transcriptomic response in the brain of spiny damselfish offspring, regardless of the duration or level of $pCO_2$ exposure. Within the control and transgenerational-elevated $pCO_2$ treatments, we found an almost 10-fold increase in the number of significantly differentially expressed genes in offspring from $CO_2$-tolerant fathers versus $CO_2$-tolerant mothers compared to the number of DEGs between the control and transgenerational-elevated $pCO_2$ treatments. A notable response uniquely upregulated in offspring of tolerant mothers was circadian rhythm regulation. This included circadian repressors that were previously found upregulated in offspring of tolerant parents when both parents exhibited tolerance in the same condition [35]. Circadian rhythm genes are closely linked to homeostatic ion-regulatory adjustments and have been identified as an important component in the response of fish to elevated $pCO_2$ [35,47–49]. Previous research linked this circadian rhythm shift to adaptive potential in the face of OA that could be inherited across a generation [35]. The ability of offspring from tolerant mothers to make changes in the circadian clock could lead to greater flexibility of ion-control and thus avoid maladaptive behavioural reactions to elevated $pCO_2$. Research on parental effects on circadian regulation especially in fish are lacking; however, several studies in mice have confirmed both the role of parental identity and maternal effects on circadian rhythm regulation of their offspring [50,51]. For example, Borengasser *et al*. [50] identified the effects of maternal obesity on circadian disruptions via suppression of key genes leading to detrimental metabolic effects in the offspring. This is consistent with our findings that the mother's phenotypic identity influences circadian rhythm regulation in juvenile fish exposed to elevated $CO_2$.

While there is an effect of parental identity on gene expression profiles with the transgenerational and control exposures, this identity weakens under acute exposure to elevated $pCO_2$. The decrease in significant differential gene expression between parental phenotypes is likely to be due to an intense cellular response to the acute exposure to elevated $pCO_2$ [52]. This population-wide response could minimize the observed molecular variation based on parental phenotype. Importantly, we found a common response to acute exposure to elevated $pCO_2$, regardless of parental condition or phenotype, which consisted of the upregulation of GABA receptor activity and neurotransmitter receptor pathways. GABA receptors play a pivotal role in maintaining the inhibitory–excitatory balance in the brain; under elevated $CO_2$ these receptors are thought to become overwhelmed by altered neuronal gradients of $Cl^-$ and $HCO_3-$ attempting to maintain pH homeostasis [20]. A recent study [21] highlights the impact of near-future $pCO_2$ levels on the creation of a vicious cycle caused by functional alterations to GABA receptors and ultimately leads to the behavioural impairments observed in fish [18,24,36]. This vicious cycle can quickly cause increased excitatory ion fluctuations in the presence of relatively small changes in $pCO_2$ levels, such as those experienced in the acute treatments. Elevated $CO_2$ could potentially have other neurological effects. For example, glycine receptors are also $Cl^-$/$HCO_3-$ channels and thus could potentially be affected by elevated $CO_2$ in a similar way to GABA receptors [53], although this has not been tested.

Indeed, we found several glycine receptors significantly upregulated in both acute-elevated $pCO_2$ conditions and offspring of both parental phenotypes. CA genes, another key player in acid-base regulation, were also identified as upregulated in both acute-elevated $pCO_2$ conditions when compared to control, regardless of parental phenotype. CAs are highly abundant in neurons and play an important role in the hydration of $CO_2$ in the cell [7,54,55]. Upregulation of these genes suggests an elevation of intracellular $HCO_3^-$ that would further excite the GABA receptors and contribute to the vicious cycle, thereby leading to an ion imbalance in the brain. Previous studies only identified CA genes to be downregulated in fishes; however, these studies focused on the gill tissues which could have different regulatory mechanisms in response to elevated $pCO_2$ than the brain [56,57]. Identification of these upregulated genes in all offspring regardless of parental phenotype suggests tolerance or sensitivity to elevated $pCO_2$ cannot protect these fish from this vicious cycle under short-term exposures. However, the absence of differential expression of these genes and pathways in offspring exposed transgenerationally to elevated $pCO_2$ suggests that acclimation via stabilization of acid-base regulation occurs with long-term exposure [58].

Welch & Munday [28] found that heritability of behavioural tolerance to increased $pCO_2$ in the fish from this experiment was only detectable in the acute exposure conditions; therefore, we endeavoured to find a molecular response correlated with the observed behavioural response between offspring of different parental pairs in the sampled acute treatments. Upregulation of genes and pathways associated with neuron plasticity and development, as well as glutamate-related genes were identified in offspring of tolerant mothers (S♂T♀). Glutamate is a key part of GABA synthesis, as well as other neurotransmitter pathways, and upregulation of these genes could be part of a compensatory mechanism along with neuronal plasticity in the face of short-term increases in $pCO_2$ [18,24,36]. The identification of downregulated growth-associated genes suggests a negative impact of elevated $pCO_2$ on development in juvenile fish and is consistent with previous research [7,59,60]. This suggests that the maternal line could be primarily responsible for neural plasticity used to compensate for changes in seawater chemistry.

In offspring of tolerant fathers (T♂S♀), we identified upregulated genes and pathways directly related to neuronal plasticity, which suggests these offspring also have compensatory neural mechanisms in response to acute increases in $pCO_2$. Histones and transcriptional repression mechanisms were found to be downregulated in this study, and in the previous study by Schunter *et al*. [36] that investigated offspring of fully tolerant parents, under acute-elevated $pCO_2$ conditions. Histones control chromatin dynamics and also play a part in gene expression by regulating transcription [61]. Decreases in repression of transcription and histone binding lead to increases in gene expression, suggesting both a genetic and epigenetic control of parental phenotype on the offspring's expression profile in response to acutely elevated $pCO_2$. Epigenetic changes can be heritable through the germ line and can be strongly influenced by environmental factors that induce specific phenotypes in parents and their offspring [62,63]. Parental influence on histone modifications are understudied; however, numerous studies have examined heritability of epigenetic changes via methylation [62,64,65].

A recent study on medaka identified paternally influenced changes in methylation patterns across multiple generations of male fish after initial exposure to hypoxia [62], whereas studies on zebrafish suggest that maternal methylation is reprogrammed in the embryo [66]. Therefore, it is possible that epigenetic influence on the offspring is more likely to come from fathers than from mothers [62,66]. Previous research on *A. polyacanthus* has already shown that epigenetics plays a key role in transgenerational acclimation to increased temperatures via histone regulation and selective DNA methylation [67,68]. This leads us to speculate an important role for epigenetic inheritance from fathers in acclimation to elevated $pCO_2$.

By combining a multi-generational experimental design with genome-wide gene expression measurements, we were able to identify effects of parental phenotypes on expression patterns of their offspring to end-of-century $pCO_2$ levels. Our results suggest that there are both maternal and paternal contributions involved in OA tolerance and that their relative importance may differ depending on length of exposure to elevated $CO_2$. Although maternal and paternal influence on specific traits is a growing research field, few studies have attempted to distinguish the separate parental contributions in transgenerational experiments [69]. In the marine stickleback, maternal influences were shown in growth rates and genetic covariance under elevated temperatures [32], but in wild salmon a correlation between telomere length and paternal growth was identified [33]. Therefore, it appears that parental contributions may vary between species as well as between specific traits. This is supported by our results; for example, maternal tolerance impacted circadian rhythm genes in their offspring while epigenetic modifications were influenced in the tolerant paternal line. The role of epigenetics in transgenerational acclimation to future climate change scenarios is also seen in recent research on DNA methylation of *A. polyacanthus* [67,68].

Further research is necessary to determine the exact mechanisms of maternal and paternal effects. Transgenerational studies spanning further generations could give insight into whether inheritance of $pCO_2$ tolerance is maintained across multiple generations, as well as clarifying the developmental time point when these expression changes occur. Also, more complex experimental designs could help clarify the relative importance of genetic versus non-genetic effects and tease apart the maternal versus paternal components of these different mechanisms. Nevertheless, it is clear from this study that variations in parental phenotype can be highly important in shaping the response of fish to future OA conditions.

Data accessibility. RNAseq data for all individuals can be found under the BioProject ID PRJNA690095. All other data are provided in the electronic supplementary material [70].

Authors' contributions. A.A.M. was involved in formal analysis, investigation, software, validation, visualization and writing the original draft; C.S. was involved in conceptualization, data curation, methodology, writing the review and editing; M.J.W. was involved in conceptualization, data curation, methodology, writing the review and editing; P.L.M.: conceptualization, funding acquisition, methodology, writing the review and editing; T.R.: conceptualization, funding acquisition, investigation, supervision, writing the review and editing. All authors gave final approval for publication and agreed to be held accountable for the work performed therein.

Competing interests. We declare we have no competing interests.

Funding. The authors acknowledge the support of the Office of Competitive Research Funds OSR-2015-CRG4–2541 from the King Abdullah University of Science and Technology (T.R. and P.L.M.), the Australian Research Council (ARC) and the ARC Centre of Excellence for Coral Reef Studies (P.L.M).

Acknowledgements. We would like to thank the KAUST Bioscience Core Lab, the KAUST integrative systems lab, the Marine Climate Change Unit at OIST, and the Marine and Aquaculture Research Facilities Unit at JCU for their help and support. We also thank R. Lehman for his help with bioinformatic analysis.

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
