## [Peer Review File · Proceedings of the Royal Society B: Biological Sciences]

Review History

RSPB-2021-0159.R0 (Original submission)

Review form: Reviewer 1

Recommendation

Accept with minor revision (please list in comments)

Scientific importance: Is the manuscript an original and important contribution to its field?

Excellent

General interest: Is the paper of sufficient general interest?

Good

Quality of the paper: Is the overall quality of the paper suitable?

Excellent

Is the length of the paper justified?

Yes

Should the paper be seen by a specialist statistical reviewer?

No

Do you have any concerns about statistical analyses in this paper? If so, please specify them explicitly in your report.

No

It is a condition of publication that authors make their supporting data, code and materials available - either as supplementary material or hosted in an external repository. Please rate, if applicable, the supporting data on the following criteria.

Is it accessible?

Yes

Is it clear?

Yes

Is it adequate?

Yes

Do you have any ethical concerns with this paper?

No

Comments to the Author

This is a well written and well organized manuscript that builds on a series of transgenerational studies from this group, each obviously involving an extraordinary amount of experimental work, from raising the fish, to sequencing and analysing the data. I find the conclusions sound and generally supportive of previous studies, but with the very novel approach of separating maternal and paternal contributions to high-CO₂ tolerance in fish, revealing some striking differences. I have only some minor comments and suggestions

line 30. Since this is a gene expression study it is a bit strong to state in the abstract that "the maternal line having greater effect of neuron growth while paternal influence impacted synaptic development." Better to state something like "expression of genes involved in....." since the study has not measured neuron growth or synaptic development.

line 75. Here it is said that "The presumption that parental care dictates the majority of parental effect has meant that paternal effects have received less attention". This cannot be true for fish since in those fishes that show parental care it is most common that it is carried out by the father, like in the stickleback mentioned in the preceding sentence.

lines 105-110. I feel that these sentences are a bit repetitive and could be condensed.

Lines 119-120. I would like to know a bit more about how the selection for T and S parents was done. Of all fish tested how many fell into the T and S categories and how many fell outside. What was the normal response to high-CO₂ in control water? Was it always <30% in CAC?

Review form: Reviewer 2

Recommendation

Major revision is needed (please make suggestions in comments)

Scientific importance: Is the manuscript an original and important contribution to its field?

Acceptable

General interest: Is the paper of sufficient general interest?

Good

Quality of the paper: Is the overall quality of the paper suitable?

Good

Is the length of the paper justified?

Yes

Should the paper be seen by a specialist statistical reviewer?

No

Do you have any concerns about statistical analyses in this paper? If so, please specify them explicitly in your report.

No

It is a condition of publication that authors make their supporting data, code and materials available - either as supplementary material or hosted in an external repository. Please rate, if applicable, the supporting data on the following criteria.

Is it accessible?

Yes

Is it clear?

Yes

Is it adequate?

No

Do you have any ethical concerns with this paper?

No

Comments to the Author

In the paper "Molecular basis of parental contributions to the behavioural tolerance of elevated pCO₂ in a coral reef fish" the authors investigate the implications of inherited resilience towards elevated CO₂ in spiny damselfish. For this, the authors assess the role of parent sex and parental phenotype, as well as the impact of acute vs chronic exposure to elevated CO₂, on gene expression in the brain of juvenile damsel fish. The authors identify a number of significant differences in the effects of maternal and paternal contribution to gene expression, as well as proposed implications and molecular mechanisms to CO₂ resilience, which could have interesting implications from both an evolutionary and ecological point of view.

The manuscript has been written in a clear and accessible manner. The introduction is well structured and the results well described. I enjoyed reading the manuscript. The topic of climate change adaptation across generations is of significant wider interest to a broad readership, whilst the investigation of parental sex and phenotype is highly novel. The manuscript therefore has the potential to make a significant contribution to the field and have significant impact. Nonetheless, there are a number of significant concerns that need addressing before the manuscript can be published in Proceedings of the Royal Society B.

General comments:

My main concern with the paper as it is currently presented is that it aims to address heritability of tolerance to elevated CO₂, setting out to assess the impact of both parental sex and parental phenotype. The title and abstract suggest the manuscript tests parental influence on behavioural pCO₂ tolerance, however at no point within this manuscript is the juvenile phenotype assessed in response to elevated CO₂. This is a real limitation of the manuscript, and significantly affects its

subsequent impact. With a number of the DEG linked to growth, at the very least I expected to see the presentation of offspring survival, length and growth data to demonstrate whether these transcriptional changes are impacting the phenotype. Moreover, as the study aims to assess behavioural tolerance, I was very surprised to see no assessment of the juvenile behavioural performance. As this was the metric measured in the parents it would have enabled an assessment of heritability of this trait. At present this is not possible. This data, if available, needs to be added, or if not available the messages throughout need to be toned down significantly. The GO results are presented as evidence of transgenerational inheritance of tolerance mechanisms, via contrasting transcriptional and epigenetic mechanisms from tolerant mothers and fathers. In the absence of any measured tolerance, the paper needs to make it far more explicit that these are simply gene expression patterns that pertain to possible tolerance, without a true assessment.

Another area that requires revision is the detail provided in the methods. At present there is a complete lack of carbonate chemistry data presented, which is not appropriate. Additionally, far more detail and clarity is required when describing and presenting the numbers of parental crosses used, as well as the number of different offspring from each parental cross. Finally, in the absence of tolerance data, I suggest adding a paragraph on future considerations, discussing the need for transgenerational studies that use more generations to really be able to disentangle the maternal/paternal/transgenerational effects on inheritance of resilience. This should also discuss the fact that we don't know if there was any ongoing selection or epigenomic processes acting in this inherited CO₂ resilience, and that investigating these points is crucial for the understanding the potential of this species to respond (either adaptive or plastic) to climate change.

Specific comments:

Abstract:

Line 28: There is no conclusive evidence that fathers affected juveniles epigenetically, so I suggest 'possible epigenetic modification' is re-written as alterations in expression of genes associated with histone binding.

Line 29: No juvenile tolerance was measured, so it is not appropriate to say parental phenotype contributed to signatures of tolerance via gene expression.

Introduction:

Line 41-42: Would it make more sense to explicitly say "acclimation potential?" as this term refers to short-term responses to environmental changes (resulting in transcriptomic variation and the aim of this paper - contrary to adaptation - long-term processes which require fixed modifications in the DNA structure - not the aim of the paper).

Line 46: This study also fails to assess heritable phenotypic variation.

Line 67 - Exemplify the non-genetic effects from the parental environment

Line 69 - Which molecular mechanisms? There are so many....

Line 73 - Maternal effects influence growth under elevated temperature. This sentence poorly describes the study results, the introduction would benefit from specifying what maternal effects the study was talking about (adaptation to higher temperatures?!) and how these effects influence offspring growth?

Line 76 - Again, linked how? This sentence needs to be re-written to properly describe the correlation!?

Material and Methods:

Line 118 – Maybe would be nice to specify which CAC was used, how it was generated?

Line 122 – Need more detail describing how many pairs per treatment, numbers of offspring produced per pair (as well as subsequently used for juvenile experiments), and volume, number and stocking density of offspring tanks. Also no data is currently presented on animal husbandry (feeding, water exchange, assessment of water quality).

Line 156 – states 80% alignment to reference genome, despite being reference genome of this species. That is considerably low. Do the authors have any ideas why there was such bad alignment? From previous studies, this is similar to alignment using a separate, but related, species genome.

Results

No data is presented on reproductive output by different pairs, or on time to laying/hatching. Was this identical across treatments, or different? Was there any implication of numbers of offspring from susceptible mothers in elevated CO₂? These data, as well as survival and growth of larvae/juveniles, in addition to juvenile behaviour, would add significant context to the study.

Line 178 – Sentence could be re-written to make it clearer. Indicate that DEGs were higher when comparing both acute to control, than when comparing transgenerational to control.

Line 214 – start the sentence with the written number

Discussion:

Line 254: could refer to figure 3 again.

Line 260: Need to be very careful in applying the term adaptation. Adaptation refers to a modification in the DNA sequence that becomes fixed in a population. More appropriate to use the term acclimation here.

Line 301: The authors state that heritability of behavioural tolerance was only noticed acutely, however in the present study any parental impact on tolerance was dwarfed by the impact of such an acute exposure on gene expression. Therefore, it would be interesting to have read more about what implications the authors felt parental tolerance mechanisms would have in real world scenarios?

Also, if behavioural tolerance is only measurable in an acute exposure, is it repeatable and persistent over time in repeated acute exposures? Could this just be a transient measure in these adults at the time of measurement, which could have shifted after the three month acclimation prior to spawning?

Whilst a clear change in gene expression is measured, it is not clear if these DEG are caused by modifications in the epigenome, or are maternal/paternal effects that could have impacted gametes and consequently the offspring performance. To truly understand if these transgenerational effects are maintained over generations, we would need to investigate those in a subsequent (third) generation. Care needs to be taken in making such strong assumptions without empirical support (this applies to the application of these terms throughout the manuscript). Maybe it would be worth adding a small paragraph about this in the discussion.

Figures:

In the figure captions, please add an explanation for what the different fish colours (e.g. blue green fish) or colours in general stand for, this will facilitate the understanding for the reader.

Decision letter (RSPB-2021-0159.R0)

01-Mar-2021

Dear Dr Monroe:

I am writing to inform you that your manuscript RSPB-2021-0159 entitled "Molecular basis of parental contributions to the behavioral tolerance of elevated pCO₂ in a coral reef fish" has, in its current form, been rejected for publication in Proceedings B.

This action has been taken on the advice of referees, who have recommended that substantial revisions are necessary. With this in mind we would be happy to consider a resubmission, provided the comments of the referees are fully addressed. However please note that this is not a provisional acceptance.

Sincerely,
Professor Gary Carvalho
mailto: proceedingsb@royalsociety.org

Associate Editor
Board Member: 1
Comments to Author:
Dear Dr Munroe

Your manuscript has been assessed by two referees who agree that it is interesting and could be published in Proc B pending some changes. Most notably of those is the request to include data on juvenile phenotypes or significantly down tone the strength and breadth of conclusions. If the requested changes can be addressed the manuscript may again be considered for publication here. All the best with your revision, Regards Line K Bay

Reviewer(s)' Comments to Author:

Referee: 1

Comments to the Author(s)

This is a well written and well organized manuscript that builds on a series of transgenerational studies from this group, each obviously involving an extraordinary amount of experimental work, from raising the fish, to sequencing and analysing the data. I find the conclusions sound and generally supportive of previous studies, but with the very novel approach of separating maternal and paternal contributions to high-CO₂ tolerance in fish, revealing some striking differences. I have only some minor comments and suggestions

line 30. Since this is a gene expression study it is a bit strong to state in the abstract that "the maternal line having greater effect of neuron growth while paternal influence impacted synaptic development." Better to state something like "expression of genes involved in....." since the study has not measured neuron growth or synaptic development.

line 75. Here it is said that "The presumption that parental care dictates the majority of parental effect has meant that paternal effects have received less attention". This cannot be true for fish since in those fishes that show parental care it is most common that it is carried out by the father, like in the stickleback mentioned in the preceding sentence.

lines 105-110. I feel that these sentences are a bit repetitive and could be condensed.

Lines 119-120. I would like to know a bit more about how the selection for T and S parents was done. Of all fish tested how many fell into the T and S categories and how many fell outside. What was the normal response to high-CO₂ in control water? Was it always <30% in CAC?

Referee: 2

Comments to the Author(s)

In the paper "Molecular basis of parental contributions to the behavioural tolerance of elevated pCO₂ in a coral reef fish" the authors investigate the implications of inherited resilience towards elevated CO₂ in spiny damselfish. For this, the authors assess the role of parent sex and parental phenotype, as well as the impact of acute vs chronic exposure to elevated CO₂, on gene expression in the brain of juvenile damsel fish. The authors identify a number of significant differences in the effects of maternal and paternal contribution to gene expression, as well as proposed implications and molecular mechanisms to CO₂ resilience, which could have interesting implications from both an evolutionary and ecological point of view.

The manuscript has been written in a clear and accessible manner. The introduction is well structured and the results well described. I enjoyed reading the manuscript. The topic of climate change adaptation across generations is of significant wider interest to a broad readership, whilst the investigation of parental sex and phenotype is highly novel. The manuscript therefore has the potential to make a significant contribution to the field and have significant impact. Nonetheless, there are a number of significant concerns that need addressing before the manuscript can be published in Proceedings of the Royal Society B.

General comments:

My main concern with the paper as it is currently presented is that it aims to address heritability of tolerance to elevated CO₂, setting out to assess the impact of both parental sex and parental phenotype. The title and abstract suggest the manuscript tests parental influence on behavioural pCO₂ tolerance, however at no point within this manuscript is the juvenile phenotype assessed in response to elevated CO₂. This is a real limitation of the manuscript, and significantly affects its subsequent impact. With a number of the DEG linked to growth, at the very least I expected to see the presentation of offspring survival, length and growth data to demonstrate whether these transcriptional changes are impacting the phenotype. Moreover, as the study aims to assess behavioural tolerance, I was very surprised to see no assessment of the juvenile behavioural

performance. As this was the metric measured in the parents it would have enabled an assessment of heritability of this trait. At present this is not possible. This data, if available, needs to be added, or if not available the messages throughout need to be toned down significantly. The GO results are presented as evidence of transgenerational inheritance of tolerance mechanisms, via contrasting transcriptional and epigenetic mechanisms from tolerant mothers and fathers. In the absence of any measured tolerance, the paper needs to make it far more explicit that these are simply gene expression patterns that pertain to possible tolerance, without a true assessment. Another area that requires revision is the detail provided in the methods. At present there is a complete lack of carbonate chemistry data presented, which is not appropriate. Additionally, far more detail and clarity is required when describing and presenting the numbers of parental crosses used, as well as the number of different offspring from each parental cross. Finally, in the absence of tolerance data, I suggest adding a paragraph on future considerations, discussing the need for transgenerational studies that use more generations to really be able to disentangle the maternal/paternal/transgenerational effects on inheritance of resilience. This should also discuss the fact that we don't know if there was any ongoing selection or epigenomic processes acting in this inherited CO₂ resilience, and that investigating these points is crucial for the understanding the potential of this species to respond (either adaptive or plastic) to climate change.

Specific comments:

Abstract:

Line 28: There is no conclusive evidence that fathers affected juveniles epigenetically, so I suggest 'possible epigenetic modification' is re-written as alterations in expression of genes associated with histone binding.

Line 29: No juvenile tolerance was measured, so it is not appropriate to say parental phenotype contributed to signatures of tolerance via gene expression.

Introduction:

Line 41-42: Would it make more sense to explicitly say "acclimation potential?" as this term refers to short-term responses to environmental changes (resulting in transcriptomic variation and the aim of this paper - contrary to adaptation - long-term processes which require fixed modifications in the DNA structure - not the aim of the paper).

Line 46: This study also fails to assess heritable phenotypic variation.

Line 67 - Exemplify the non-genetic effects from the parental environment

Line 69 - Which molecular mechanisms? There are so many....

Line 73 - Maternal effects influence growth under elevated temperature. This sentence poorly describes the study results, the introduction would benefit from specifying what maternal effects the study was talking about (adaptation to higher temperatures?!) and how these effects influence offspring growth?

Line 76 - Again, linked how? This sentence needs to be re-written to properly describe the correlation!?

Material and Methods:

Line 118 - Maybe would be nice to specify which CAC was used, how it was generated?

Line 122 - Need more detail describing how many pairs per treatment, numbers of offspring produced per pair (as well as subsequently used for juvenile experiments), and volume, number

and stocking density of offspring tanks. Also no data is currently presented on animal husbandry (feeding, water exchange, assessment of water quality).

Line 156 – states 80% alignment to reference genome, despite being reference genome of this species. That is considerably low. Do the authors have any ideas why there was such bad alignment? From previous studies, this is similar to alignment using a separate, but related, species genome.

Results

No data is presented on reproductive output by different pairs, or on time to laying/hatching. Was this identical across treatments, or different? Was there any implication of numbers of offspring from susceptible mothers in elevated CO₂? These data, as well as survival and growth of larvae/juveniles, in addition to juvenile behaviour, would add significant context to the study.

Line 178 – Sentence could be re-written to make it clearer. Indicate that DEGs were higher when comparing both acute to control, than when comparing transgenerational to control.

Line 214 – start the sentence with the written number

Discussion:

Line 254: could refer to figure 3 again.

Line 260: Need to be very careful in applying the term adaptation. Adaptation refers to a modification in the DNA sequence that becomes fixed in a population. More appropriate to use the term acclimation here.

Line 301: The authors state that heritability of behavioural tolerance was only noticed acutely, however in the present study any parental impact on tolerance was dwarfed by the impact of such an acute exposure on gene expression. Therefore, it would be interesting to have read more about what implications the authors felt parental tolerance mechanisms would have in real world scenarios?

Also, if behavioural tolerance is only measurable in an acute exposure, is it repeatable and persistent over time in repeated acute exposures? Could this just be a transient measure in these adults at the time of measurement, which could have shifted after the three month acclimation prior to spawning?

Whilst a clear change in gene expression is measured, it is not clear if these DEG are caused by modifications in the epigenome, or are maternal/paternal effects that could have impacted gametes and consequently the offspring performance. To truly understand if these transgenerational effects are maintained over generations, we would need to investigate those in a subsequent (third) generation. Care needs to be taken in making such strong assumptions without empirical support (this applies to the application of these terms throughout the manuscript). Maybe it would be worth adding a small paragraph about this in the discussion.

Figures:

In the figure captions, please add an explanation for what the different fish colours (e.g. blue green fish) or colours in general stand for, this will facilitate the understanding for the reader.

Author's Response to Decision Letter for (RSPB-2021-0159.R0)

See Appendix A.

RSPB-2021-1931.R0 (Original submission)

Review form: Reviewer 1

Recommendation

Accept with minor revision (please list in comments)

Scientific importance: Is the manuscript an original and important contribution to its field?

Excellent

General interest: Is the paper of sufficient general interest?

Good

Quality of the paper: Is the overall quality of the paper suitable?

Excellent

Is the length of the paper justified?

Yes

Should the paper be seen by a specialist statistical reviewer?

No

Do you have any concerns about statistical analyses in this paper? If so, please specify them explicitly in your report.

No

It is a condition of publication that authors make their supporting data, code and materials available - either as supplementary material or hosted in an external repository. Please rate, if applicable, the supporting data on the following criteria.

Is it accessible?

Yes

Is it clear?

Yes

Is it adequate?

Yes

Do you have any ethical concerns with this paper?

No

Comments to the Author

I am pleased to find that the authors have responded and edited the manuscript appropriately along the lines of my suggestions and I have therefore no further comments.

Review form: Reviewer 3

Recommendation

Accept with minor revision (please list in comments)

Scientific importance: Is the manuscript an original and important contribution to its field?

Excellent

General interest: Is the paper of sufficient general interest?

Excellent

Quality of the paper: Is the overall quality of the paper suitable?

Excellent

Is the length of the paper justified?

Yes

Should the paper be seen by a specialist statistical reviewer?

No

Do you have any concerns about statistical analyses in this paper? If so, please specify them explicitly in your report.

No

It is a condition of publication that authors make their supporting data, code and materials available - either as supplementary material or hosted in an external repository. Please rate, if applicable, the supporting data on the following criteria.

Is it accessible?

Yes

Is it clear?

Yes

Is it adequate?

No

Do you have any ethical concerns with this paper?

No

Comments to the Author

Overall this is an interesting paper looking at the effects of parental contributions on the responses of clownfish to elevated pCO₂ in the context of climate change. The authors have done a good job of revising their manuscript and addressing the reviewers' comments. I have a couple of minor additional comments that should be addressed.

Line numbers refer to the revised manuscript.

Line 190: the second "and" should be changed to "an".

Lines 302 - 309: Another plausible explanation is an overall change in neuronal activity in the brain under elevated pCO₂ consistent with changes in behaviour. The upregulation in GABA receptors activity could simply be a compensatory mechanism to combat potential changes in glutamate release in the brain (as suggested by Tresguerres and Hamilton 2017, p 2141). No measurements of actual chloride and bicarbonate concentrations have been performed in the

neurons and these are inferred from changes in these ions in the plasma or whole-brain homogenates. Even slight changes in the parameters used to calculate GABAA receptor equilibrium could influence whether these are depolarizing or hyperpolarizing (again see Tresguerres and Hamilton 2017, p 2143-2144). Therefore, other potential explanations should be mentioned here.

SupplementaryInfo file:

Lines 6-8: How much time passes between the make-up of CAC and behavioural testing? CAC degrades very quickly (half-life of fewer than 30 minutes) so must be made and used fresh. Please clarify.

Decision letter (RSPB-2021-1931.R0)

08-Nov-2021

Dear Dr Monroe

I am pleased to inform you that your manuscript RSPB-2021-1931 entitled "Molecular basis of parental contributions to the behavioral tolerance of elevated $p\text{CO}_2$ in a coral reef fish" has been accepted for publication in Proceedings B.

The referee(s) have recommended publication, but also suggest some minor revisions to your manuscript. Therefore, I invite you to respond to the referee(s)' comments and revise your manuscript. Because the schedule for publication is very tight, it is a condition of publication that you submit the revised version of your manuscript within 7 days. If you do not think you will be able to meet this date please let us know.

- 1) A text file of the manuscript (doc, txt, rtf or tex), including the references, tables (including captions) and figure captions. Please remove any tracked changes from the text before submission. PDF files are not an accepted format for the "Main Document".
- 2) A separate electronic file of each figure (tiff, EPS or print-quality PDF preferred). The format should be produced directly from original creation package, or original software format. PowerPoint files are not accepted.
- 3) Electronic supplementary material: this should be contained in a separate file and where possible, all ESM should be combined into a single file. All supplementary materials accompanying an accepted article will be treated as in their final form. They will be published

alongside the paper on the journal website and posted on the online figshare repository. Files on figshare will be made available approximately one week before the accompanying article so that the supplementary material can be attributed a unique DOI.

If you wish to submit your data to Dryad (<http://datadryad.org/>) and have not already done so you can submit your data via this link [http://datadryad.org/submit?journalID=RSPB&manu=\(Document not available\)](http://datadryad.org/submit?journalID=RSPB&manu=(Document not available)) which will take you to your unique entry in the Dryad repository. If you have already submitted your data to dryad you can make any necessary revisions to your dataset by following the above link. Please see <https://royalsociety.org/journals/ethics-policies/data-sharing-mining/> for more details.

Sincerely,
Professor Gary Carvalho
<mailto:proceedingsb@royalsociety.org>

Reviewer(s)' Comments to Author:

Referee: 1

Comments to the Author(s).

I am pleased to find that the authors have responded and edited the manuscript appropriately along the lines of my suggestions and I have therefore no further comments.

Referee: 3

Comments to the Author(s).

Overall this is an interesting paper looking at the effects of parental contributions on the responses of clownfish to elevated pCO₂ in the context of climate change. The authors have done a good job of revising their manuscript and addressing the reviewers' comments. I have a couple of minor additional comments that should be addressed.

Line numbers refer to the revised manuscript.

Line 190: the second "and" should be changed to "an".

Lines 302 - 309: Another plausible explanation is an overall change in neuronal activity in the brain under elevated pCO₂ consistent with changes in behaviour. The upregulation in GABA receptors activity could simply be a compensatory mechanism to combat potential changes in glutamate release in the brain (as suggested by Tresguerres and Hamilton 2017, p 2141). No measurements of actual chloride and bicarbonate concentrations have been performed in the neurons and these are inferred from changes in these ions in the plasma or whole-brain homogenates. Even slight changes in the parameters used to calculate GABA_A receptor equilibrium could influence whether these are depolarizing or hyperpolarizing (again see Tresguerres and Hamilton 2017, p 2143-2144). Therefore, other potential explanations should be mentioned here.

SupplementaryInfo file:

Lines 6-8: How much time passes between the make-up of CAC and behavioural testing? CAC degrades very quickly (half-life of fewer than 30 minutes) so must be made and used fresh. Please clarify.

Author's Response to Decision Letter for (RSPB-2021-1931.R0)

See Appendix B.

Decision letter (RSPB-2021-1931.R1)

15-Nov-2021

Dear Dr Monroe

I am pleased to inform you that your manuscript entitled "Molecular basis of parental contributions to the behavioral tolerance of elevated pCO₂ in a coral reef fish" has been accepted for publication in Proceedings B.

Data Accessibility section

Open Access

Paper charges

Sincerely,

Proceedings B

Appendix A

RSPB-2021-0159

Molecular basis of parental contributions to the behavioral tolerance of elevated pCO₂ in a coral reef fish

Referee: 1

Comments to the Author(s)

This is a well written and well-organized manuscript that builds on a series of transgenerational studies from this group, each obviously involving an extraordinary amount of experimental work, from raising the fish, to sequencing and analyzing the data. I find the conclusions sound and generally supportive of previous studies, but with the very novel approach of separating maternal and paternal contributions to high-CO₂ tolerance in fish, revealing some striking differences. I have only some minor comments and suggestions

Thank you for reviewing our manuscript. We appreciate your comments and have addressed all your suggestions. Please find the details below.

line 30. Since this is a gene expression study it is a bit strong to state in the abstract that "the maternal line having greater effect of neuron growth while paternal influence impacted synaptic development." Better to state something like "expression of genes involved in....." since the study has not measured neuron growth or synaptic development.

Sentence has been rewritten as suggested (line 30).

line 75. Here it is said that "The presumption that parental care dictates the majority of parental effect has meant that paternal effects have received less attention". This cannot be true for fish since in those fishes that show parental care it is most common that it is carried out by the father, like in the stickleback mentioned in the preceding sentence.

Good point. We were referring to paternal effects more broadly here but as this paragraph is all about fish, we agree it doesn't fit. The sentence has been removed.

lines 105-110. I feel that these sentences are a bit repetitive and could be condensed.

Thanks for pointing this out. The sentences have been edited to convey the information in a clearer way (line 133).

Lines 119-120. I would like to know a bit more about how the selection for T and S parents was done. Of all fish tested how many fell into the T and S categories and how many fell outside. What was the normal response to CAC in control water? Was it always <30% in CAC?

Adults were only tested in elevated pCO₂ conditions. We have added extra information on this data into the supplementary information file, lines 15-20. 38.02% of collected adults fell into the tolerant category while 38.84% fell into the defined sensitive category. In the previous study offspring that were tested in control conditions spent an average time of <30% in the CAC (1).

Referee: 2

Comments to the Author(s)

In the paper “Molecular basis of parental contributions to the behavioural tolerance of elevated pCO₂ in a coral reef fish” the authors investigate the implications of inherited resilience towards elevated CO₂ in spiny damselfish. For this, the authors assess the role of parent sex and parental phenotype, as well as the impact of acute vs chronic exposure to elevated CO₂, on gene expression in the brain of juvenile damsel fish. The authors identify a number of significant differences in the effects of maternal and paternal contribution to gene expression, as well as proposed implications and molecular mechanisms to CO₂ resilience, which could have interesting implications from both an evolutionary and ecological point of view.

The manuscript has been written in a clear and accessible manner. The introduction is well structured and the results well described. I enjoyed reading the manuscript. The topic of climate change adaptation across generations is of significant wider interest to a broad readership, whilst the investigation of parental sex and phenotype is highly novel. The manuscript therefore has the potential to make a significant contribution to the field and have significant impact. Nonetheless, there are a number of significant concerns that need addressing before the manuscript can be published in Proceedings of the Royal Society B.

Thank you for taking the time to review our paper. We appreciate your efforts and are happy you enjoyed the manuscript.

General comments:

My main concern with the paper as it is currently presented is that it aims to address heritability of tolerance to elevated CO₂, setting out to assess the impact of both parental sex and parental phenotype. The title and abstract suggest the manuscript tests parental influence on behavioural pCO₂ tolerance, however at no point within this manuscript is the juvenile phenotype assessed in response to elevated CO₂. This is a real limitation of the manuscript, and significantly affects its subsequent impact. With a number of the DEG linked to growth, at the very least I expected to see the presentation of offspring survival, length and growth data to demonstrate whether these transcriptional changes are impacting the phenotype. Moreover, as the study aims to assess behavioural tolerance, I was very surprised to see no assessment of the juvenile behavioural performance. As this was the metric measured in the parents it would have enabled an assessment of heritability of this trait. At present this is not possible. This data, if available, needs to be added, or if not available the messages throughout need to be toned down significantly. The GO results are presented as evidence of transgenerational inheritance of tolerance mechanisms, via contrasting transcriptional and epigenetic mechanisms from tolerant mothers and fathers. In the absence of any measured tolerance, the paper needs to make it far more explicit that these are simply gene expression patterns that pertain to possible tolerance, without a true assessment.

Another area that requires revision is the detail provided in the methods. At present there is a complete lack of carbonate chemistry data presented, which is not appropriate. Additionally, far more detail and clarity is required when describing and presenting the numbers of parental crosses used, as well as the number of different offspring from each parental cross.

Finally, in the absence of tolerance data, I suggest adding a paragraph on future considerations, discussing the need for transgenerational studies that use more generations to really be able to disentangle the maternal/paternal/transgenerational effects on inheritance of resilience. This should also discuss the fact that we don't know if there was any ongoing selection or epigenomic processes acting in this inherited CO₂ resilience, and that investigating these points is crucial for the understanding the potential of this species to respond (either adaptive or plastic) to climate change.

You have raised important concerns here. We have described the changes we made to the manuscript both in reference to these overall comments and the specific ones below. The heritability of this phenotype has already been examined and published in previous research. Welch and Munday (1) measured the behavioural responses to CAC of both parents and offspring at elevated pCO₂ in this experiment and estimated heritability using parent-offspring regression. We have clarified this in the materials and methods section (lines 142-144) as well as by adding further information on the methods in the supplementary information file. Due to size constraints (to obtain sufficient brain tissue for transcriptomic analysis), we were unable to use exactly the same juvenile fish to measure transcriptomic data as had been used to compare behaviour in Welch and Munday (1). Those fish were sampled after 6 weeks while we grew out our fish to 5 months to ensure sufficient amounts of brain tissue. However, the fish are siblings and come from the same experiment. This information has been added into the methods at lines 179 to 194 to help clarify the differences between the two studies using the same experiment. We also chose not to behaviorally phenotype these fish prior to euthanasia to avoid any effects of disturbance from behavioural testing on the transcriptome. The previous study by Welch and Munday identified heritable phenotypic variation when offspring were exposed acutely to elevated pCO₂, but not when they were exposed chronically for 6 weeks (1). This is mentioned in the discussion starting at line 381 where we explore the molecular responses that could be responsible for this heritable phenotype under acute exposure to elevated pCO₂.

Welch and Munday reported the behavioral data for this experiment, but not other traits, such as survival and size. We have added observations of survival (lines 165,174) as well as life history data in the form of body weight when these fish were euthanized. This raw data can be found in Supplementary data 1 (Table 3) as well as statistical analysis between parental pairs and conditions in the new supplementary information file, which includes a boxplot as supplementary figure 1. This revealed an interesting result that matched the differentially expressed growth genes we identified in tolerant mothers. Thank you for this suggestion as it added a new aspect to the data that we hadn't considered.

We have also added further information in the methods (line 162, 177), as well as in the new supplementary information document to include the missing data and detail. This includes a section on the water chemistry (Supplementary table 1) and the CAC process. Lines in the methods clarify the number of pairs of adults used as well as the total offspring samples per parental pair, per condition.

As recommended, we have also added a few sentences to the end of the manuscript describing future considerations and how this study could be expanded to tease apart maternal and

paternal contributions. This along with our clarifications regarding the previous study focused on heritable behavior should clarify the scope and aims of our study.

Specific comments:

Abstract:

Line 28: There is no conclusive evidence that fathers affected juveniles epigenetically, so I suggest 'possible epigenetic modification' is re-written as alterations in expression of genes associated with histone binding.

Sentence has been modified as suggested (line 28).

Line 29: No juvenile tolerance was measured, so it is not appropriate to say parental phenotype contributed to signatures of tolerance via gene expression.

Sentence has been rewritten to better state the results (line 28-29).

Introduction:

Line 41-42: Would it make more sense to explicitly say "acclimation potential?" as this term refers to short-term responses to environmental changes (resulting in transcriptomic variation and the aim of this paper - contrary to adaptation – long-term processes which require fixed modifications in the DNA structure – not the aim of the paper).

Acclimation potential fits here, but so does adaptive. We have decided to include both words in the sentence at line 51. This paper is looking at transcriptomic variation but in the context of heritability based on results of the previous study examining behavior.

Line 46: This study also fails to assess heritable phenotypic variation.

Again, we are assessing transcriptomic variation of a behavior that has already been shown to be heritable. However, based on the recommendation we have removed heritable from the line 55 to better match our specific study.

Line 67 – Exemplify the non-genetic effects from the parental environment

We have changed the wording throughout this paragraph to clarify the non-genetic and genetic effects discussed. We have also added in further details on the examples of non-genetic parental effects from lines 83 to 90.

Line 69 – Which molecular mechanisms? There are so many....

We discuss these in the preceding and following paragraphs that are most relevant to our study. This includes disruption to neurotransmitters such as GABA_A (lines 62-64), shifts in circadian rhythm, and changes to their acid-base regulation (lines 118-122). We also altered a couple of the following sentences in this paragraph to discuss specific mechanisms as

mentioned below (lines 83-90).

Line 73 – Maternal effects influence growth under elevated temperature. This sentence poorly describes the study results, the introduction would benefit from specifying what maternal effects the study was talking about (adaptation to higher temperatures?!) and how these effects influence offspring growth?

Sentence has been updated to better describe the results and included the mechanisms behind these maternal effects in lines 83-87.

Line 76 – Again, linked how? This sentence needs to be re-written to properly describe the correlation!?

This sentence has been rewritten to better explain the results in lines 87-90, again with associated mechanisms.

Material and Methods:

Line 118 – Maybe would be nice to specify which CAC was used, how it was generated?

Thanks for the suggestion. This information has been added to the supplementary information from lines 3 to 13.

Line 122 – Need more detail describing how many pairs per treatment, numbers of offspring produced per pair (as well as subsequently used for juvenile experiments), and volume, number and stocking density of offspring tanks. Also no data is currently presented on animal husbandry (feeding, water exchange, assessment of water quality).

Most of this information has been added to the supplementary information (lines 23-36). Water quality information can be found in Supplementary Table 1. We also updated several sentences in the material and methods section to help clarify the number of pairs and offspring used in this experiment (lines 163-164). This data can also be found in the file supplementary data 1 (Table 2).

Line 156 – states 80% alignment to reference genome, despite being reference genome of this species. That is considerably low. Do the authors have any ideas why there was such bad alignment? From previous studies, this is similar to alignment using a separate, but related, species genome.

The fish used in this study are from a different population on the Great Barrier Reef than those individuals used to create the reference genome. Some possible SNPs could be the cause for this lower alignment. We also used very strict parameters to ensure the best data quality which could have reduced this as well. When using the same species reference genome, alignment usually falls between 80-90% and we are within that range.

Results

No data is presented on reproductive output by different pairs, or on time to laying/hatching. Was this identical across treatments, or different? Was there any implication of numbers of offspring from susceptible mothers in elevated CO₂? These data, as well as survival and growth of larvae/juveniles, in addition to juvenile behaviour, would add significant context to the study.

No impact of elevated pCO₂ on reproductive output was observed in this experiment and no significant juvenile mortality occurred. This information has been added into the manuscript at lines 165 and 174 respectively. Based on this and your earlier suggestions we have added growth data as body weight at time of euthanasia into the supplementary information (lines 42-58, supplementary figure 1). We discovered a significant difference in body weight between offspring of different parental phenotypes under transgenerational exposure to elevated pCO₂. Overall the average total weight was higher in offspring of tolerant fathers than tolerant mothers across all conditions. This finding of downregulated growth and development genes in tolerant mother offspring combined with upregulation of genes related to neural plasticity suggest a possible trade-off of body size when compensating for changes in pCO₂. We have also added this into the results at lines 266-269. Thank you for the suggestion.

Line 178 – Sentence could be re-written to make it clearer. Indicate that DEGs were higher when comparing both acute to control, than when comparing transgenerational to control.

Thank you for pointing this out. The sentence has been edited to make the results clearer (lines 235-237).

Line 214 – start the sentence with the written number

Thanks for catching this. The number has been written out (line 280).

Discussion:

Line 254: could refer to figure 3 again.

Thank you for the suggestion but we have decided not to add the figure reference at what is now line 322 in the discussion.

Line 260: Need to be very careful in applying the term adaptation. Adaptation refers to a modification in the DNA sequence that becomes fixed in a population. More appropriate to use the term acclimation here.

At this point in the discussion (line 328) adaptation is the correct wording. We are discussing previous research that examined the possible heritability of tolerance from parents to offspring, linked to changes in circadian rhythm (2).

Line 301: The authors state that heritability of behavioural tolerance was only noticed acutely, however in the present study any parental impact on tolerance was dwarfed by the impact of such

an acute exposure on gene expression. Therefore, it would be interesting to have read more about what implications the authors felt parental tolerance mechanisms would have in real world scenarios?

This is a great point. Our study shows that possible genetic mechanisms of parental tolerance appear to be inherited even in long-term exposures. However, the phenotypic variation disappears suggesting nonadaptive plasticity in these fish under chronic high pCO₂. This is definitely something to be explored in further studies as we have now mentioned in the final paragraph (Lines 435-439). Perhaps adding a third generation while measuring both transcriptomic and phenotypic variation could help us determine if phenotypic variation continues to diminish over time and whether any genetic signatures are impacted.

Also, if behavioural tolerance is only measurable in an acute exposure, is it repeatable and persistent over time in repeated acute exposures? Could this just be a transient measure in these adults at the time of measurement, which could have shifted after the three month acclimation prior to spawning?

Welch and Munday (1) tested the repeatability of behavioural responses to CAC in adult fish exposed to elevated and found that it was highly repeatable (R squared 0.99) after 24 hours. Tolerant adults stayed tolerant and sensitive adults stayed sensitive over this timeframe. However, repeatability over the longer term has not been tested. It is entirely possible that behavioural tolerance to elevated CO₂ observed in acute treatments with adults declines over time when permanently exposed to elevated CO₂, as observed in juveniles, meaning that all adults would respond equally in chronic elevated CO₂ treatment. This would make it extremely difficult to test for heritability using parent-offspring correlations, because there would be little observed variation among adults.

Whilst a clear change in gene expression is measured, it is not clear if these DEG are caused by modifications in the epigenome, or are maternal/paternal effects that could have impacted gametes and consequently the offspring performance. To truly understand if these transgenerational effects are maintained over generations, we would need to investigate those in a subsequent (third) generation. Care needs to be taken in making such strong assumptions without empirical support (this applies to the application of these terms throughout the manuscript). Maybe it would be worth adding a small paragraph about this in the discussion.

Thanks for this suggestion. A paragraph describing the usefulness of future research in teasing apart these effects has been added to the end at lines 435-439.

Figures:

In the figure captions, please add an explanation for what the different fish colours (e.g. blue green fish) or colours in general stand for, this will facilitate the understanding for the reader.

We have added an explanation into the Figure 1 caption and expanded the Figure 3 caption to clarify the color use.

References

1. Welch MJ, Munday PL. Heritability of behavioural tolerance to high CO₂ in a coral reef fish is masked by nonadaptive phenotypic plasticity. *Evol Appl.* 2017;
2. Schunter C, Welch MJ, Ryu T, Zhang H, Berumen ML, Nilsson GE, et al. Molecular signatures of transgenerational response to ocean acidification in a species of reef fish. *Nat Clim Chang.* 2016;6(11):1014–8.

Appendix B

Referee: 1

Comments to the Author(s).

I am pleased to find that the authors have responded and edited the manuscript appropriately along the lines of my suggestions and I have therefore no further comments.

Thank you for reviewing our manuscript again. We appreciate the time and effort you have put in and our manuscript has been made better for it.

Referee: 3

Comments to the Author(s).

Overall this is an interesting paper looking at the effects of parental contributions on the responses of clownfish to elevated pCO₂ in the context of climate change. The authors have done a good job of revising their manuscript and addressing the reviewers' comments. I have a couple of minor additional comments that should be addressed.

Thank you for taking the time to review our manuscript. We appreciate your comments and have indicated below our revisions based on your suggestions.

Line numbers refer to the revised manuscript.

Line 190: the second "and" should be changed to "an".

Thank you for catching this error. We have fixed it in the manuscript at line 190.

Lines 302 - 309: Another plausible explanation is an overall change in neuronal activity in the brain under elevated pCO₂ consistent with changes in behaviour. The upregulation in GABA receptors activity could simply be a compensatory mechanism to combat potential changes in glutamate release in the brain (as suggested by Tresguerres and Hamilton 2017, p 2141). No measurements of actual chloride and bicarbonate concentrations have been performed in the neurons and these are inferred from changes in these ions in the plasma or whole-brain homogenates. Even slight changes in the parameters used to calculate GABAA receptor equilibrium could influence whether these are depolarizing or hyperpolarizing (again see Tresguerres and Hamilton 2017, p 2143-2144). Therefore, other potential explanations should be mentioned here.

Thanks for this suggestion. We have added additional information regarding impacts to neuronal activity under elevated pCO₂ to this paragraph at lines 310-315. Although we didn't use the glutamate example from the suggested paper, we were interested in their suggestion about the impacts on glycine receptors. We found some of these genes upregulated in our own data set and have include this information in the discussion.

SupplementaryInfo file:

Lines 6-8: How much time passes between the make-up of CAC and behavioural testing? CAC

degrades very quickly (half-life of fewer than 30 minutes) so must be made and used fresh.
Please clarify.

We have added several sentences from lines 9-11 in the supplementary information file to clarify that fresh CAC was prepared for each trial and used within 10 minutes of preparation.